# In Vitro Hormetic Effect Investigation of Thymol on Human Fibroblast and Gastric Adenocarcinoma Cells

**DOI:** 10.3390/molecules25143270

**Published:** 2020-07-17

**Authors:** Ayse Günes-Bayir, Abdurrahim Kocyigit, Eray Metin Guler, Agnes Dadak

**Affiliations:** 1Department of Nutrition and Dietetics, Faculty of Health Sciences, Bezmialem Vakif University, Silahtaraga Cad., Eyüpsultan 34065, Istanbul, Turkey; 2Department of Medical Biochemistry, Faculty of Medicine, Bezmialem Vakif University, Vatan Cad., Fatih 34093, Istanbul, Turkey; akocyigit@bezmialem.edu.tr (A.K.); emguler@bezmialem.edu.tr (E.M.G.); 3Institute of Pharmacology, Department for Biomedical Sciences, University of Veterinary Medicine Vienna, Veterinärplatz 1, A-1210 Vienna, Austria; Agnes.Dadak@vetmeduni.ac.at

**Keywords:** thymol, hormetic effect, cancerous cells, healthy cells

## Abstract

The concept of hormesis includes a biphasic cellular dose-response to a xenobiotic stimulus defined by low dose beneficial and high dose inhibitory or toxic effects. In the present study, an attempt has been made to help elucidate the beneficial and detrimental effects of thymol on different cell types by evaluating and comparing the impact of various thymol doses on cancerous (AGS) and healthy (WS-1) cells. Cytotoxic, genotoxic, and apoptotic effects, as well as levels of reactive oxygen species and glutathione were studied in both cell lines exposed to thymol (0–600 µM) for 24 h. The results showed significant differences in cell viability of AGS compared to WS-1 cells exposed to thymol. The differences observed were statistically significant at all doses applied (*P* ≤ 0.001) and revealed hormetic thymol effects on WS-1 cells, whereas toxic effects on AGS cells were detectable at all thymol concentrations. Thymol at low concentrations provides antioxidative protection to WS-1 cells in vitro while already inducing toxic effects in AGS cells. In that sense, the findings of the present study suggest that thymol exerts a dose-dependent hormetic impact on different cell types, thereby providing crucial information for future in vivo studies investigating the therapeutic potential of thymol.

## 1. Introduction

The term of hormesis has been widely utilized in the field of biomedical, nutrition and toxicological sciences [1]. The hormetic effect of a substance is described as a biphasic dose-response to an environmental or chemical agent with a low dose stimulation or favorable effect and a high dose inhibitory or toxic effect on the cells or organisms. In a review article, it was reported that a great number of anticancer agents proliferate human cancer cells at low doses based on the hormetic dose-response relationship [2]. In general, natural phenolic compounds have been gaining increasing attention in cancer chemoprevention and anticancer treatment [3]. It has been shown that polyphenols inhibit cell proliferation and induce apoptosis in vitro and in vivo [3,4]. There are two main pathways; the intrinsic and extrinsic pathways that have been linked to phyto-compound induced apoptosis. The intrinsic pathway is mainly activated by the release of the Bcl-2 family proteins from the mitochondria and the extrinsic pathway by the activation of transmembrane death receptors and ligands [5]. Studies provided evidence for the redox status of glutathione (GSH) playing a critical role in cell apoptosis in various cell types [6,7,8]. It has been revealed that most anticancer or cancer-chemopreventive agents have the potential to induce apoptosis. Although these apoptotic effects are significant for cancer chemoprevention, the cytotoxic and genotoxic activity of such agents in healthy cells are unwanted. Therefore, an ideal anticancer agent should increase apoptosis and inhibit the proliferation of cancer cells while minimally affecting healthy cells.

Thymol, as a phenolic compound is known to possess therapeutic qualities such as antioxidative, antimicrobial, antiinflammatory, and anticancer activities [9]. Therefore, thymol has gained more and more scientific attention in recent years. The cytotoxic, genotoxic, and antioxidative potentials of thymol have been investigated in vitro on various healthy cells: human lymphocytes, V79 Chinese hamster lung fibroblasts, mouse cortical neurons and peripheral-blood mononuclear cells [10,11,12,13]. Additionally, the anticancer effect of thymol has been studied in in vitro systems. Various cell lines such as human glioblastoma, promyelocytic cancer, gastric adenocarcinoma, colon carcinoma, hepatoma and lung carcinoma cells [11,14,15,16,17,18,19,20] were used, but the clinical significance of the thymol-induced effects remains unknown. To the authors knowledge, only a couple in vivo studies are published, one reporting that 100 mg of thymol per kg of body weight induced genotoxic effects on bone marrow cells in rats [21]. Another study focused on in vivo passage kinetics and has evidenced thymol metabolization in the stomach or intestine after an oral application [22]. Our research recently revealed that human gastric adenocarcinoma (AGS) cells are highly sensitive to thymol in a concentration-dependent manner [14,15]. Given those studies, it is of great interest to elucidate in depth whether thymol may have therapeutic potential against e.g., gastric cancer. On the other hand, the hormetic effect of isothiocyanates depends on the investigated cell types which are healthy or cancer cells [23]. Therefore, we focused first on an in vitro comparison of the proposed thymol activity on human gastric adenocarcinoma (AGS) cells with findings gained from thymol-treated healthy human fibroblasts (WS-1) at different concentrations. The study was designed to gain a more in-depth insight into the thymol-sensitivity of various cell lines in order to disclose potential limitations of this phenolic compound as a natural anticancer drug aspirant.

## 2. Results

### 2.1. Cell Viability

The inhibitory effect of thymol on cell viability was assessed in both cell cultures using the ATP cell viability assay. Thymol showed cytotoxic effects by reducing cell viability on AGS and WS-1 cells in a dose-dependent manner (Figure 1). Statistically significant differences were found among thymol (50–600 μM) exposed healthy and cancerous cells (*P* ≤ 0.001). However, thymol at the lowest concentration used (10 µM) did not significantly increase the number of healthy cells when it was compared to untreated control cells. In addition, IC_50_ values of thymol were 75.63 ± 4.01 μM and 167± 11 μM in cancerous and healthy cells, respectively.

### 2.2. Reactive Oxygen Species (ROS)

The pro-oxidative effect of thymol was studied using the oxidation-sensitive fluorescent dye DCFH-DA. A dose-dependent intracellular reactive oxygen species (ROS) generating effect of thymol was detected in cells. Differences were statistically significant among thymol (20–100 μM) exposed healthy and cancerous cells (*P* ≤ 0.001) (Figure 2). An increased thymol concentration and relative ROS levels in AGS cells were positively correlated. However, no statistically significant differences were observed in WS-1 cells treated with thymol (10–100 μM) as compared to ROS levels of control cells.

### 2.3. Glutathione (GSH) Level

In order to assess the antioxidative effect of thymol, GSH levels were measured using GSH/GSSG-Glo assay. Thymol (20–100 µM) induced a dose-dependent reduction in GSH levels in healthy and cancerous cell lines. An increased concentration of thymol in the range of 20–100 µM and the relative GSH level in both cell types were negatively correlated. At 20 µM, GSH levels of AGS cells were significantly higher than in WS-1 cells (*P* ≤ 0.001) (Figure 3). On the other hand, GSH levels in both cell lines at 20–100 µM of thymol were significantly reduced in comparison with their controls (*P* ≤ 0.001). At the lowest concentration used (10 µM), healthy cells indicated the same GSH level as unexposed control cells, whereas GSH levels started to decrease in AGS cells exposed to the same concentration.

### 2.4. Effect of Thymol on Apoptosis Induction

To assess the cytotoxic effect of thymol, whether caused by apoptosis or not, the AO/EB staining was applied to visualize nuclear changes and apoptosis-characteristic body formation. After staining, the cells were observed under a fluorescence microscope and counted to quantify apoptosis. Cell morphology was determined after exposing to thymol (0–100 μM) for 24 h. In both cell lines, the cell viability was decreased significantly at 10–50 µM (*P* ≤ 0.001). The number of apoptotic cells at 10–20 µM and 50–100 µM increased significantly in a dose-dependent manner (*P* ≤ 0.001) (Figure 4a). A significant difference was also detected for necrotic cells at the highest dose (100 µM) (*P* ≤ 0.001) (Figure 4b).

An expressional analysis of apoptosis proteins was examined in healthy and cancerous cells by Western blot analysis in order to evaluate whether thymol induces apoptosis via caspase activation. Changes were observed in both cell lines when cells were treated with different doses of thymol. Protein expressions for both cell lines are demonstrated in Figure 5a–d. *β-actin* was used as a control, and was positive in all samples. The results revealed that the expression levels of *Bax, Caspase-9, and Caspase-3* proteins were significantly increased in both cell lines exposed to 50 μM thymol whereas *Bcl-2* protein was decreased in a dose-dependent manner (Figure 5a–d; *P* ≤ 0.001). It was also suggested that thymol induces apoptosis via decreasing the *Bcl-2/Bax* ratio and overexpression of *Caspase-9* and *Caspase-3* proteins. However, *Bcl-2* levels in healthy cells exposed to thymol (30 and 50 μM) were higher than in cancerous cells (*P* ≤ 0.001).

### 2.5. DNA Damage

The comet assay was used for the determination of thymol induced DNA damage in this study. Genotoxic effects of thymol on both cell lines were visualized as comet formation. The percentages of DNA in the tail are shown for both cell lines (Figure 6). A positive correlation was detected among the DNA damage and an increased dose of thymol in both cell lines. Differences in the amount of tail DNA were detected statistically significant among healthy and cancerous cells (*P* ≤ 0.001).

## 3. Discussion

Phytochemicals are gaining increasing attention as tentative chemopreventive or anticancer agents. Various natural compounds have been investigated for the use in various types of cancers using in vitro and in vivo models [3]. It seems that polyphenols may have the potential to express anticancer activity by inhibiting the multiplication of cancerous cells and stimulating apoptosis. However, the clinical efficacy and safety of these phytocompounds needs to be carefully evaluated in terms of avoiding toxicological side effects on healthy cells [24]. The hormetic effect of an agent, defined as a biphasic dose-response to an agent characterized by a beneficial effect at low doses and a toxic/inhibitory activity at high doses, depends on the selected endpoint and/or the healthy or cancer cells studied [3]. The present study revealed that thymol exhibits a hormetic dose-response in the sense of antioxidant and pro-oxidant activity on cell viability and DNA genotoxicity. The observed low dose benefit of thymol might be beneficial for cancer chemoprevention whilst the data on its high dose effects provide valuable information for future in vivo studies investigating the therapeutic anticancer potential of this phytocompound. Thymol is one of the major compounds found in members of the *Lamiaceae* family [9] and some other plants. This natural monoterpene phenol has important biological activities; however, its effect on cancer has not yet been fully elucidated [15]. In the present study, the cytotoxic, apoptotic, genotoxic, pro-oxidative and antioxidative effects of thymol were investigated in healthy and cancerous cell lines, and the results from both cell lines were compared. As evident from the present results, thymol induces intracellular ROS generation with the concomitant onset of cytotoxicity, apoptosis, and genotoxicity. The effects on human gastric adenocarcinoma cells were more pronounced than on healthy human fibroblast cells. These findings are in accordance with a report on peripheral blood mononuclear cells (PBMC) and acute promyelocytic cancer cells [11].

In the present study, the hormetic effect of thymol was examined using ATP cell viability and AO/EB staining assays, respectively. Cancerous and healthy cells exposed to thymol for 24 h were evaluated for apoptotic, necrotic, and viable cells by apoptosis assay. The impaired cell functions induced by thymol were in a dose-dependent manner and more pronounced in cancerous cells than healthy cells. To our knowledge, only one study has compared the thymol effect on the cell viability and apoptosis in cancerous (acute promyelocytic cancer HL-60) and healthy (peripheral blood mononuclear) cells [11] so far, and found similar results. Thymol-induced suppression of cell proliferation was observed in a dose-dependent manner in various cancer cell types such as human gastric adenocarcinoma AGS, hepatocellular carcinoma HepG2, lung carcinoma H1299, or glioblastoma cells [15,16,19,20]. However, studies on human colon carcinoma Caco-2 cells and primary cultures of mouse cortical neuron cells exposed to thymol (0–250 µM and 1–1000 µM, respectively) showed no cytotoxicity; but ultrastructural changes and induction of apoptosis were evident in Caco-2 cells [12,18].

Phenolic compounds, such as thymol, are suggested to display either antioxidative or prooxidant activity depending on concentration, cell resistance, and exposing time [25]. In this regard, the cell altering mechanism of thymol has been linked to its capacity of disrupting cytoplasmic membranes, especially mitochondrial membranes, resulting in a prooxidant status and subsequent apoptosis induction [13,19,26]. Recent studies reported that thymol at lower doses protects human colon carcinoma, glioblastoma, acute promyelocytic leukemia and gastric adenocarcinoma cells against ROS, whereas higher doses induce oxidative stress by increasing ROS generation [11,15,16,17,26]. A study performed on fibroblast cells showed a slight reduction in ROS levels when using thymol at a concentration of 1–100 µM [13]. Although there is essential research data characterizing the apoptotic and antioxidative effects of thymol, limited information is available on its prooxidant activity. In this respect, it is valuable to monitor intracellular levels of reduced glutathione (GSH) which has been shown to be crucial for the regulation of cell proliferation, cell cycle progression, and apoptosis [27]. The main function of GSH is to reduce intracellularly numerous oxidizing compounds, including ROS. With respect to the potential therapeutic use of thymol as a (co)drug in cancer treatment, the effect of thymol on intracellular ROS and GSH levels was investigated at different doses in vitro in AGS versus WS-1 cells. A positive correlation was found among cell death and ROS production after 24 h of exposure to thymol at higher doses, whereas a negative correlation was observed between cell death and GSH levels. These findings are in agreement with earlier reports [11,13,14,15]. It is noteworthy to mention that in the present study, low doses of thymol induced the increased proliferation of healthy human fibroblasts as compared to unexposed control cells. This activity was accompanied by unaltered ROS and decreased GSH levels in healthy cells, whereas none of these effects were seen in cancerous cells, which were harmed by thymol even at the lowest concentration. This result might be of special value when selecting doses for in vivo studies.

Apoptosis as the gene-directed and programmed cell death [5] is regulated by different proteins such as *Bax*, *Bcl-2*, *Caspase-3,* and *Caspase-9*. In the present study, the apoptotic effect of thymol (0–50 µM) was investigated at the protein expression level in cancerous and healthy cells. Protein levels of *Bcl-2* decreased in both thymol-treated (0–50 μM) cell lines, but *Bcl-2* levels were significantly higher in healthy cells than in cancerous cells. Similarly, one study reported that thymol (0–50 μM) induced apoptosis in human promyelocytic leukemia (HL-60) cells in a concentration-dependent manner, involving both caspase-dependent and caspase-independent pathways [11]. To our knowledge, our study is the first report describing the *Caspase-3* and *Caspase-9* activity of thymol on cancerous cells in comparison to healthy cells. The down-regulation of *Bcl-2* and up-regulation of *Bax, Caspase-9,* and *Caspase-3* protein expression in cancerous cells may draw attention to thymol as a potential candidate for the development of future gastric adenocarcinomas treatment strategies. The observed effect that anti-apoptotic *Bcl-2* remained upregulated in healthy cells seems appealing in this context.

It has been suggested that phenolic compounds induce DNA damage via their pro-oxidative effects on cells [4]. In the present study, the genotoxic effect of thymol (0–50 µM) was assessed via detecting DNA damage by comet assay. Thymol (10–50 µM) showed genotoxic effects on both cell types in a dose-dependent manner, but this effect was more statistically significant in cancerous cells than healthy cells. To our knowledge, the present report is the first study evaluating and comparing genotoxic effects of thymol on cancer and healthy cells. One study was recently published on the apoptotic as well as genotoxic effects of thymol on cancerous human AGS cells [15]. However, genotoxicity tests were included in studies on other cancerous cells such as human non-small lung cancer (H1299) and human hepatoma (HepG2) cells exposed to thymol which showed data similar to our findings [19,20]. Another study reported that thymol (25 µM) induces genotoxicity in the V79 cells [13]. Additionally, a study on Caco-2 and HepG2 cells exposed to thymol up to 600 μM revealed cytotoxicity, but the results were not associated with DNA-damaging effects [28]. Studies mentioned above were conducted exclusively on cancer cells. In an in vivo experiment, thymol caused genotoxicity at all doses used (40, 60, 80, and 100 mg/kg body weight) in rat bone marrow cells. Interestingly, thymol induced numerical chromosome abnormalities at the highest concentration [21]. Further in vivo studies are required to explore whether low doses of thymol can be better tolerated by healthy cells, as observed in our in vitro experiments, while being still effective against cancerous cells.

## 4. Material and Methods

### 4.1. Chemicals

Materials purchased from Sigma Chemical Co. (St. Louis, MO, USA) were thymol, DMSO (dimethyl sulfoxide), DCFH-DA (2′,7′- dichlorofluorescein-diacetate), AO/EB (acridine orange/ethidium bromide) stain, SDS polyacrylamide gels, coomassie brilliant-blue-dye, powdered skim milk, trypan blue, comet assays, low-melting agarose, normal melting agarose, and lysis solution. Ham’s F-12 culture medium, fetal bovine serum (FBS), and antibiotics (100 U/mL penicillin, 100 μg/mL streptomycin) were obtained from Gibco Invitrogen Corporation (Carlsbad, CA, USA). CellTiter-Glo Luminescent Cell Viability and GSH/GSSG-Glo assays were provided by Promega (Madison, MI, USA). *Bax* (N-20), *Bcl-2* (C-21), *Caspase-3* (H-277), *Caspase-9* (H-170), and *β-actin* (AC-15) antibodies were purchased from Santa Cruz Biotechnologies (Santa Cruz, CA, USA). Amersham ECL Plus Western Blotting Detection reagents were obtained from GE Healthcare (Piscataway, NJ, USA). Radioimmunoprecipitation assay (RIPA) buffer and a proteinase inhibitor cocktail were from Roche (Mannheim, Germany). Molecular weight marker BenchMark Pre-Stained Protein Ladder was from Invitrogen (Grand Island, New York, USA).

### 4.2. Cell Cultures

Human cancerous AGS and healthy WS-1 cells purchased from the American Type Culture Collection (ATCC, Manassas, VA, USA) were cultured in Ham’s F-12 Medium which was supplemented with 10% FBS and antibiotics (100 U/mL penicillin, 100 μg/mL streptomycin). AGS and WS-1 cells were plated at a density of 15 × 10^3^ cells/mL in 96-well plates and at 18 × 10^4^ cells/mL in 6-well plates and incubated for 24 h at 37 °C.

### 4.3. Treatments

A dose range of thymol was prepared for the determination of the half-maximal inhibitory concentration (IC_50_) value [11,15,17]. According to these literatures, different doses of thymol (0, 10, 20, 30, 50, 100, 200, and 400 μM) were prepared from a stock solution of 600 μM of thymol in DMSO. The final concentration of DMSO in the medium was 0.1%, and was used as a negative control.

### 4.4. Cytotoxic Activity Assay

The CellTiter-Glo Luminescent Cell Viability Assay was used for the determination of the number of viable cells in the sense of identifying the cytotoxic effect of thymol in both lines. Human AGS and WS-1 cells were seeded at a density of 15 × 10^3^ cells mL^−1^ in each well of 96-well plates. After 24 h incubation, both cell lines were treated with different doses (0–600 μM) of thymol and incubated at 37 °C further. The CellTiter-Glo Reagent was added into each well, and then the ATP in cells was quantified using a luminometer (Varioskan Flash Multimode Reader, Thermo, Waltham, MA). The viability of cells was demonstrated as a percentage following comparison with the negative control. The IC_50_ value was calculated from the concentration response curves by non-linear regression analysis.

### 4.5. Prooxidant Activity Assay

Reactive oxygen species were detected and measured by oxidative-sensitive fluorescent DCFH-DA probe. Two-hundred microliters of medium and 15 × 10^3^ mL^−1^ AGS and WS-1 cells were placed into 96-well microtiter plates for their 24 h incubation. After that, the media were replaced with a growth medium containing 5% FBS. Plates were incubated with added different concentrations of thymol (0–100 μM) for another 24 h. Both cell cultures were washed with cold PBS and incubated with 100 mM DCFH-DA for 30 min at 37 °C. The fluorescence intensity was measured using the fluorescence plate reader (Varioskan Flash Multimode Reader, Thermo, Waltham, MA) at Ex./Em. = 488/525 nm). Measurements were carried out to ensure each time that the number of cells per treatment group had the same reproducibility. The results were reported as a percentage of relative fluorescence relative to the control cells.

### 4.6. Antioxidant Activity Assay

The GSH/GSSG-Glo assay was used to determine and quantify the GSH level in cells which is an indicator of oxidative stress. One hundred microliters of complete medium and both cell lines were plated at a density of 15 × 10^3^ mL^−1^ in each well of plate. Cells were treated with different doses of thymol (0–100 μM). The method was carried out according to the manufacturer’s instructions. The luminescence measurements are a quantification method based on light emission existing GSH.

### 4.7. Apoptotic Activity Assays

AO/EB fluorescent dyes were used to investigate the apoptotic activity of thymol (0–100 μM) at the cellular level. Both cell lines were plated at a density of 18 × 10^4^ mL^−1^ in each well of plates and were exposed to different doses of thymol for 24 h incubation. Further examinations were performed according to manufacturer instructions. Multiple photos were taken at randomly-selected areas, and a minimum of 100 cells were counted using fluorescence microscopy (Leica DM1000, Solms, Germany). Nuclear morphologies of cells were assessed according to the previously described method [14,29].

The Western blot method was used to determine the apoptotic activity of thymol (0–100 μM) at the protein expression level. After treatment with different doses of thymol (0–50 μM), AGS and WS-1 cells were lysed in a radioimmunoprecipitation assay (RIPA) buffer (50 mM Tris-HCl, pH 7.4, 150 mM NaCl, 5 mM EDTA, 1% Nonidet P-40, 1% sodium deoxycholate, 0.1% SDS, 1% aprotinin, 50 mM NaF, 0.1 mM Na_3_VO_4_) and a proteinase inhibitor cocktail. Lysates were centrifuged at 13.000× *g* for 15 min at +4 °C. The pellet was discarded, and the supernatant containing the protein was transferred into a clean tube. The cell protein was measured by the Bradford Coomassie brilliant blue dye method. Samples containing 30 μg of protein, together with the molecular weight marker were subjected to 10% SDS polyacrylamide gel electrophoresis under reducing conditions. The proteins were transferred to polyvinylidene difluoride membranes. They were blocked with 5% powdered skim milk in 0.05%, Triton X-100/Tris-buffered saline with Tween for 1 h at room temperature. Blots were incubated overnight with primary antibodies *Bax*, *Bcl-2*, *Caspase-3*, and *Caspase-9* for 1 h then incubated with peroxidase-conjugated goat anti-rabbit secondary antibody. All samples were also blotted for *β-actin* in order to normalize protein amounts. Amersham ECL Plus Western Blotting Detection Reagents (GE Healthcare, Piscataway, NJ, USA) were used to determine the protein expression levels which were captured with an imaging system (Vilber Lourmat Sté, Collégien, France).

### 4.8. Genotoxicity Assay

Alkaline single cell gel electrophoresis (SCGE, comet assay) was used to determine the genotoxic activity of thymol (0–50 μM) on both cell lines. Before performing the comet assay, the trypan blue exclusion test was carried out for the determination of viable cell numbers present in the cell suspensions. Human AGS and WS-1 cells were plated at a density of 2 × 10^5^ cells/well in six-well plates and incubated at 37 °C for 24 h. Different doses of thymol (0–50 μM) were added into both cell cultures. DMSO (1%) was used as a negative control, and 50 µM H_2_O_2_ was applied as a positive control. The assay was performed according to the method of Singh et al. [30] with slight modifications as described earlier [14]. Additionally, a computerized image analysis system (Comet Assay IV; Perceptive Instruments) was carried out. The percentage (%) of tail DNA was determined and used as a measure for damaged DNA [31].

### 4.9. Statistical Analyses

The obtained data were analyzed using the Statistical Package for the Social Sciences version 23.0 (SPSS Inc, Chicago, IL, USA). All the data were expressed as the mean ± SD of the number of experiments. All performed experiments were done in triplicate, and the standard deviation was within 5%. Percentages were calculated in relation to control cells. Likewise, a non-linear regression analysis was performed for the estimation of the IC_50_ value of thymol in healthy and cancer cell lines. All the data were tested using ANOVA, and post hoc analyses of different doses of thymol were performed by Tukey’s test. The means of both cell lines were compared by the Mann-Whitney-U test. Pearson’s correlation coefficient test was presented to assess the associations between ROS generation and cell viability parameters. *P* ≤ 0.001 was considered significant.

## 5. Conclusions

This study demonstrated the first in vitro comparison of a wide range of thymol concentrations in order to investigate the hormetic effect of thymol on healthy or cancer cells. The findings of the present study suggest that thymol exerts a dose-dependent hormetic impact on the different cell types. Whereas higher doses of this phenolic compound may equally harm cancerous and healthy cells, low doses seem to protect healthy cells but harm cancer cells. On the basis of the described thymol-characteristics, it seems attractive to speculate about thymol as a tentative candidate for the development of future anticancer treatment strategies. However, further studies need to provide a more detailed understanding of thymol-induced effects on cancer and healthy cells at low doses. The better understanding of hormetic mechanisms of thymol at the cellular and molecular levels can be of special value for future in vivo studies which are of utmost importance to explore whether thymol could indeed be a potential preventive and/or therapeutic option in the future.

## Figures and Tables

**Figure 1 molecules-25-03270-f001:**
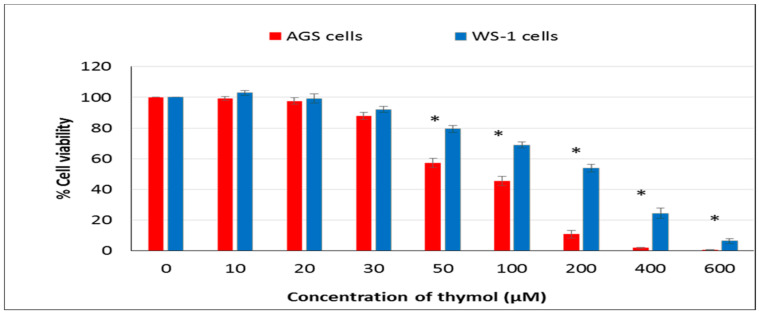
Cytotoxicity effect of thymol (0–600 μM) after 24 h incubation was studied in healthy and cancerous cells. All values are expressed as the mean ± SD. * Differences were considered significant compared to the control group from *P* ≤ 0.001. SD: Standard deviation.

**Figure 2 molecules-25-03270-f002:**
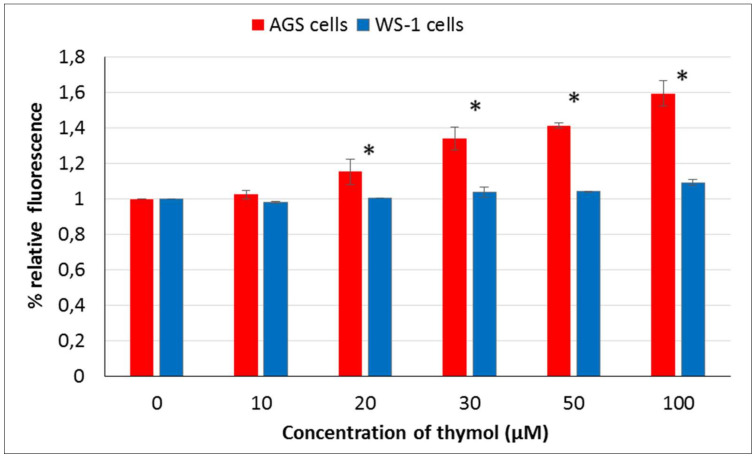
Reactive oxygen species (ROS) levels in healthy and cancerous cells exposed to thymol (0–100 μM) were investigated by DCFH-DA assay after 24 h incubation. All values are expressed as the mean ± SD. * Differences were considered significant compared to the control group from *P* ≤ 0.001. SD: Standard deviation.

**Figure 3 molecules-25-03270-f003:**
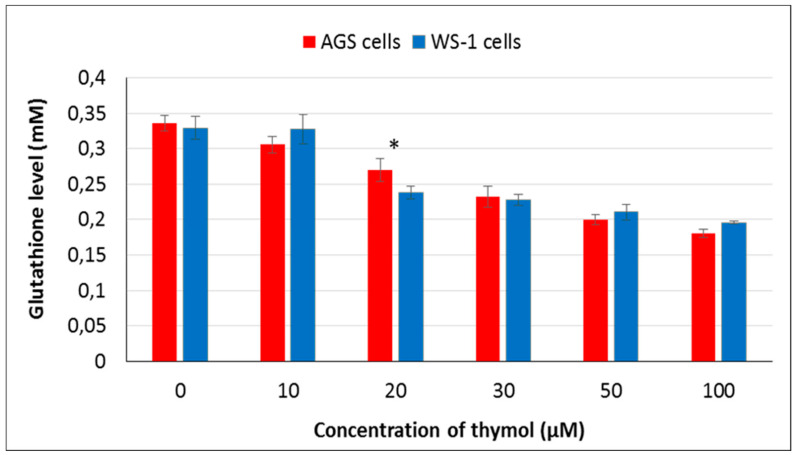
GSH levels in healthy and cancerous cells were shown after 24 h of exposure to thymol (0–100 μM). All values are expressed as the mean ± SD. * Differences were considered significant compared to the control group from *P* ≤ 0.001. SD: Standard deviation.

**Figure 4 molecules-25-03270-f004:**
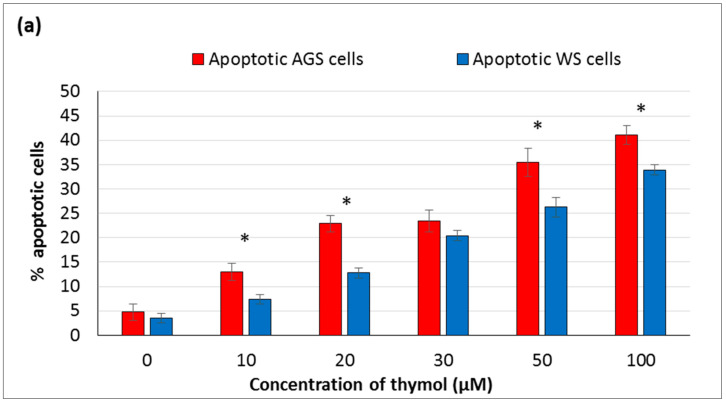
Morphological changes in healthy and cancerous cells were demonstrated, which were exposed to thymol (0–100 μM) after 24h incubation. (**a**) Apoptotic and **(b)** necrotic effects of thymol on both cell cultures are given in comparison with control cells as percentages. All values are expressed as the mean ± SD. * Differences were considered significant compared to the control group from *P* ≤ 0.001. SD: Standard deviation.

**Figure 5 molecules-25-03270-f005:**
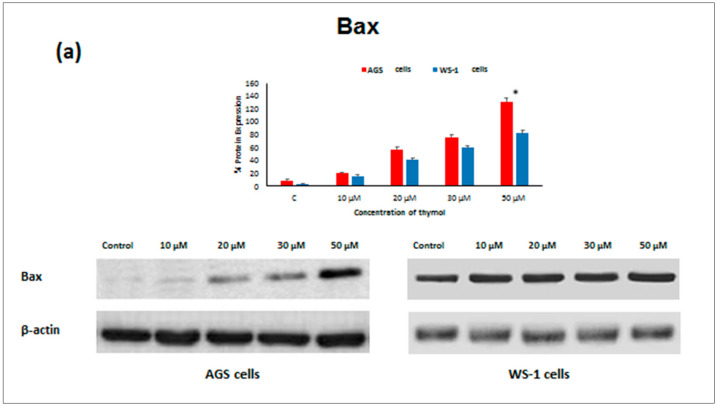
Thymol induced apoptosis was detected by Western blotting. Healthy and cancerous cells were treated with 0–50 μM of thymol. Expressions of *Bax* (**a**), *Bcl-2* (**b**), *Caspase-3* (**c**), and *Caspase-9* (**d**) proteins are presented as percentages as well as on gel. All values are expressed as the mean ± SD. * Differences were considered significant compared to the control group from *P* ≤ 0.001. SD: Standard deviation. C: Control.

**Figure 6 molecules-25-03270-f006:**
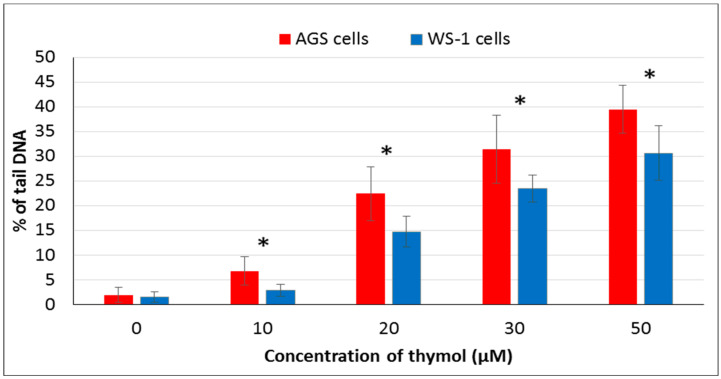
The genotoxic effect of thymol on healthy and cancerous cells was detected by comet assay. Cells exposed to thymol (0–50 μM) were analyzed after 24 h of incubation. All values are expressed as the mean ± SD. * Differences were considered significant compared to the control group from *P* ≤ 0.001. SD: Standard deviation.

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
