# Peer review of "In Vitro Hormetic Effect Investigation of Thymol on Human Fibroblast and Gastric Adenocarcinoma Cells"

_molecules, 2020, doi:10.3390/molecules25143270_

Round 1
Reviewer 1 Report
Authors presented the hormetic effect of thymol on normal and gastric adenocarcinoma cells, studying the antioxidative effects, apoptosis and necrosis of cells in a dose-dependent manner. Thymol is a phenolic compound already known for its therapeutic qualities and the goal of this study is very interesting.
There are some minor comments to take into consideration:
- The definition of the figures is very bad and for some graphs it was difficult to read the y-axis information. Please, improve the definition of all graphs.
- In line 75, authors present a remarkably increase number of healthy cells viability in the 10 uM treatment; this is an exaggerated comment as from the graph it is shown a very slight increase which seems insignificant.
- In line 88, it is not clear where this slight reduction is. Please check again the comments in lines 86-89, as, additionally, healthy cells do not show a significant increase of ROS levels in a dose-dependent manner.
- Paragraph 2.3. The differences in glutathione levels in terms of mM do not really show this significant difference between normal and cancer cells. Numerically, these differences are very low. In any case, the comment of lines 100-101 is evident for the 100um thymol treatment, as the 50uM show standard deviations of AGS and WS-1 cells that do not really explain this significant difference. Please, carefully check these data.
- Check typing errors in lines: 57 “an other” is another; 300 “previousyl” is previously; 302 “blott” is blot; 315 “for with” is for; 341 “is demonstrated” is “demonstrated”
Major comment:
- In addition to the low definition of figure 5 which makes it difficult to read the legends, there is a poor description of the result of the experiments and a mismatched graph corresponding to the blotting results. Observing the intensity western blot results of Bax, Caspase-3 and -9, it looks like their protein quantity is higher in WS-1 cells than in AGS cells, whereas the density of beta-actin in cancer cells is stronger than in normal cells, so it is not really understandable how the graphic shows e higher protein amount in cancer cells normalized to beta-actin when treated with 20 and 30 uM thymol. Please check again these results and describe them better in the “Results” (lines 127-132) and in the “Discussion” (lines 211-222).
Author Response
Dear Reviewer,
We changed and corrected our manuscript according to your proposals which have improved it.
Please see the attachment.
Thank you very much for your kindly support and effort!
Best regards,
Ayse

Reviewer 2 Report
Bayir et al. interestingly shown the Hormetic effect of the Thymol on fibroblast and adenocarcinoma cells. These finding also precisely showed the dose-dependent effect on in vitro system, which could crucial information for future studies.
I have just 2 minor comments.
1) Image seems little distorted and not visible clearly. please use high resolution images.
2) Please shade a light on a molecular mechanism via representative figure showing the impact of the different molecules that has been tested or get hampered by Thymol treatment. Also mention about possible cell signaling pathways that could involve in this treatment.
Author Response

(The authors gave the same response as above.)
